# Data-driven Reconstruction of Partially Observed Dynamical Systems

Pierre Tandeo[1,2,3], Pierre Ailliot[4], and Florian Sévellec[5,2]

[1]IMT Atlantique, Lab-STICC, UMR CNRS 6285, F-29238, France
[2]Odyssey, Inria/IMT/CNRS, France
[3]RIKEN Center for Computational Science, Kobe, 650-0047, Japan
[4]Univ Brest, UMR CNRS 6205, Laboratoire de Mathematiques de Bretagne Atlantique, France
[5]Laboratoire d'Océanographie Physique et Spatiale, Univ Brest CNRS IRD Ifremer, Brest, France

**Correspondence:** Pierre Tandeo (pierre.tandeo@imt-atlantique.fr)

**Abstract.** The state of the atmosphere, or of the ocean, cannot be exhaustively observed. Crucial parts might remain out of reach of proper monitoring. Also, defining the exact set of equations driving the atmosphere and ocean is virtually impossible because of their complexity. The goal of this paper is to obtain predictions of a partially observed dynamical system, without knowing the model equations. In this data-driven context, the article focuses on the Lorenz-63 system, where only the second and third components are observed, and access to the equations is not allowed. To account to those strong constraints, a combination of machine learning and data assimilation techniques is proposed. The key aspects are the following: the introduction of latent variables, a linear approximation of the dynamics, and a database that is updated iteratively, maximising the likelihood. We find that the latent variables inferred by the procedure are related to the successive derivatives of the observed components of the dynamical system. The method is also able to reconstruct accurately the local dynamics of the partially observed system. Overall, the proposed methodology is simple, easy to code, and gives promising results, even in the case of small amounts of observations.

## 1 Introduction

In geophysics, even if one has the perfect knowledge of the studied dynamical system, it remains difficult to predict because of the existence of nonlinear processes (Lorenz, 1963). Beyond this important difficulty, achieving this perfect knowledge of the system is often impossible. Consequently, the governing differential equations are often not known in full because of their complexity, in particular regarding scale-interactions (e.g., turbulent closures are often assumed rather than "known" *per se*). On top of these two major difficulties, the state of the system is not and cannot be exhaustively observed. Potentially crucial components are and might remain partly or fully out of reach of proper monitoring (e.g., deep ocean or small scale features). Predicting a partially observed and partially known system is therefore a key issue in current geophysics and in particular for ocean, climate and atmospheric sciences.

A typical example of such a framework is the use of climate indices (e.g., Global Mean Temperature, Niño 3.4 index, North Atlantic Oscillation index) and the study of their links and their dynamics. In this context, the direct relationship between those

indices is unknown, even if their more indirect and complex relation exist, through the full knowledge of the climate dynamics. Also, it is highly possible that climate indices are dependent on components of the climate that are not currently considered as key indices, and so are not fully monitored. However, these key indices could be sufficient to describe the most important aspect of climate, leading to accurate and reliable predictions, and enabling cost-effective adaptation and mitigation.

Hence, an alternative to physics-based models is to use available observations of the system and statistical approaches to discover equations, and then make predictions. This has been introduced in several papers, using combinations and polynomials of observed variables, as well as sparse regressions or model selection strategies (Brunton et al., 2016; Rudy et al., 2017; Mangiarotti and Huc, 2019). Those methods have then been extended to the case of noisy and irregular observation sampling, using a Bayesian framework as in data assimilation (Bocquet et al., 2019; North et al., 2022). Alternatively, some authors used data assimilation and local linear regressions based on analogs (Tandeo et al., 2015; Lguensat et al., 2017), or iterative data assimilation coupled with neural networks (Brajard et al., 2020; Fablet et al., 2021; Brajard et al., 2021), to make data-driven predictions without discovering equations.

However, many approaches cited above are assuming that the full state of the system is observed, which is a strong assumption. Indeed, in a lot of applications in geophysics, important components of the system are never or only partially observed such as the deep ocean (see e.g., Jayne et al., 2017), and data-driven methods fail to make good predictions. To deal with those strong constraints, i.e., when the model is unknown and when some components of the system are never observed, combination of data assimilation and machine learning shows potential (see e.g., Wikner et al., 2021). Additionally, an option is to use time-delay embedding of the available components of the system (Takens, 1981; Brunton et al., 2017), whereas another option is to find latent representations of the dynamical system (see e.g., Talmon et al., 2015; Ouala et al., 2020). In this study, we will show that they are strong relationships between those two approaches.

Here, we propose a simple algorithm using linear and Gaussian assumptions, based on a state-space formulation. This classic Bayesian framework, used in data assimilation, is able to deal with a dynamical model (physics- or data-driven) and observations (partial and noisy). Three main ideas are used: (i) augmented state formulation (Kitagawa, 1998), (ii) global linear approximation of the dynamical system (Korda and Mezić, 2018), and (iii) estimation of the dynamical parameters using an iterative algorithm combined with Kalman recursions (Shumway and Stoffer, 1982). The current paper is thus an extension of (Shumway and Stoffer, 1982) to never observed components of a dynamical system, using a state augmentation strategy. The proposed framework is probabilistic, where the state of the system is approximated using a Gaussian distribution (with a mean vector and a covariance matrix). The algorithm is iterative, where a catalog is updated at each iteration and used to learn a linear dynamical model. The final estimate of this catalog corresponds to a new system of variables, including latent ones.

The proposed methodology is based on an important assumption: the surrogate model is linear. Although it can be considered as a disadvantage compared to nonlinear models, this linear assumption also has interesting properties. Indeed, nonlinear model combined with state-augmentation is a very broad family of model and may lead to identifiability issues. Using a linear dynamics already leads to a very flexible family of model since the latent variable may describe nonlinearities and include for example any transformation of the observed or non-observed components of a dynamical model. Furthermore, it allows a rigorous estimation of the parameters using well established statistical algorithms which can be run at a low computational

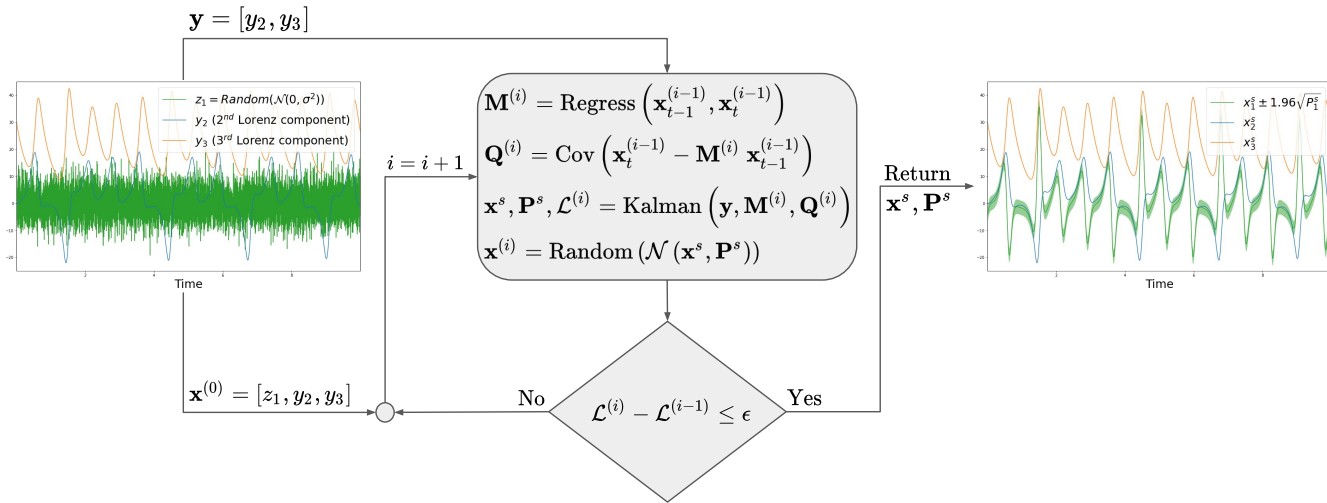

**Figure 1.** Schematic of the proposed methodology, illustrated using the Lorenz-63 system. The algorithm is initialized with a Gaussian random noise for the hidden component (i.e., $z_1$) and with partial observations of the system (i.e., $y_2$ and $y_3$). Then, an iterative procedure is applied with a linear regression, a covariance computation, the Kalman recursions, and a random sampling. This algorithm is iteratively maximizing the likelihood of the observations noted $\mathcal{L}$. After convergence of the algorithm, a hidden component $z_1$ is stabilized and represented by a Gaussian distribution represented by the mean $x_1^s$ and variance $P_1^s$.

cost. The proposed methodology is evaluated on a low-dimensional and weakly nonlinear chaotic model. As this paper is a proof of concept, a linear surrogate model is certainly well suited for this situation.

The paper is organized as follows. Firstly, the methodology is explained in section 2. Secondly, section 3 describes the experiment using the Lorenz-63 system. Thirdly, the results are reported in section 4. The conclusions and perspectives are drawn in section 5.

## 2   Methods

The methodology proposed in this paper is borrowed from data assimilation, machine learning, and dynamical systems. It is
summarized in Fig. 1 and explained below.

In data assimilation, the goal is to estimate, from partial and noisy observations $\mathbf{y}$, the full state of a system $\mathbf{x}$. When the dynamical model used to propagate $\mathbf{x}$ in time is available (i.e., when model equations are given), classic data assimilation techniques are used to retrieve unobserved components of the system. For instance, in the Lorenz-63 system (Lorenz, 1963), if only 2 variables ($x_2$ and $x_3$ in the example defined below) are observed, knowing the Lorenz equations (system of three
ordinary differential equations), it is possible to retrieve the unobserved one ($x_1$ in our example below). But this estimation requires good estimates of model and observations error statistics (see e.g., Dreano et al., 2017; Pulido et al., 2018).

Now, if the model equations are not known and observations of the system are available over a sufficient period of time, it is possible to use data-driven methods to mathematically approximate the system dynamics. In this paper, a linear approximation is used to model the relationship of the state vector $\mathbf{x}$ between two time steps. It is parameterized with the matrix $\mathbf{M}$, which dimension is equal to the square of the state-space. Moreover, a linear observation operator is introduced to relate the partial observations $\mathbf{y}$ and the state $\mathbf{x}$. It is written using a matrix $\mathbf{H}$, with its dimension equal to the observation-space times the state-space dimensions. Nonlinear and adaptive operators as well as noisy observations could be taken into account but, for the sake of simplicity, only the linear and non-noisy case is considered in this paper.

Mathematically, matrices $(\mathbf{M}, \mathbf{H})$ and vectors $(\mathbf{x}, \mathbf{y})$ are linked using a Gaussian and linear state-space model such that

$$\mathbf{x}_t = \mathbf{M}\mathbf{x}_{t-1} + \boldsymbol{\eta}_t, \tag{1a}$$

$$\mathbf{y}_t = \mathbf{H}\mathbf{x}_t + \boldsymbol{\epsilon}_t, \tag{1b}$$

where $t$ is the time index and $\boldsymbol{\eta}_t$ and $\boldsymbol{\epsilon}_t$ are unbiased Gaussian vectors, representing the model and observation errors, respectively. Their error covariance matrices are noted $\mathbf{Q}$ and $\mathbf{R}$, respectively. Those matrices indirectly control the respective weight given to the model and to the observations. It constitutes an important tuning part of the state-space models (see Tandeo et al., 2020, for a more in depth discussion).

In such a data-driven problem where only a part of the system is observed, a first natural step is to consider that the state $\mathbf{x}$ is directly related to the observations $\mathbf{y}$. For instance, in the example of the Lorenz-63 previously introduced, observations correspond to the second and third components of the system (i.e., $x_2$ and $x_3$, formally defined later).

In this paper, we propose to introduce a hidden vector noted $\mathbf{z}$, corresponding to one or more hidden components that are not observed. To this purpose, the state is augmented using this hidden component $\mathbf{z}$, the observation vector $\mathbf{y}$ does not change, and the operator $\mathbf{H}$ is a truncated identity matrix. The use of augmented state-space is classic in data assimilation and mostly refer to the estimation of unknown parameters of the dynamical model (see Ruiz et al., 2013, for further details).

The hidden vector $\mathbf{z}$ is now accounted in the linear model $\mathbf{M}$, given in Eq. (1a), whose dimension has increased. The hidden components are completely unknown and thus randomly initialized using Gaussian white noises, parameterized by $\sigma^2$, their level of variance. The next step is to infer $\mathbf{z}$ using a statistical estimation method. Starting from the random initialization, an iterative procedure is proposed, based on the maximization of the likelihood.

The proposed approach is based on a linear and Gaussian state-space model given in Eqs. (1) and thus uses the classic Kalman filter and smoother equations. The Kalman filter (forward in time) is used to get the information of the likelihood, whereas the Kalman smoother (forward and backward in time) is used to get the best estimate of the state. The proposed approach is inspired by the Expectation-Maximization algorithm (noted EM, see Shumway and Stoffer, 1982) and is able to iteratively estimate the matrices $\mathbf{M}$ and $\mathbf{Q}$. In this paper, $\mathbf{R}$ is assumed known and negligible. The criterion used to update those matrices is based on the innovations, defined by the difference between the observations $\mathbf{y}$ and the forecast of the model $\mathbf{M}$, noted $\mathbf{x}^f$. The likelihood of the innovations, noted $\mathcal{L}$, is computed using $T$ time steps such that:

$$\mathcal{L} \triangleq p\left(\mathbf{y}_1, \ldots, \mathbf{y}_T | \mathbf{x}_1^f, \ldots, \mathbf{x}_T^f\right) \propto \prod_{t=1}^{T} \exp\left(-\left(\mathbf{y}_t - \mathbf{H}\mathbf{x}_t^f\right)^\top \boldsymbol{\Sigma}_t^{-1} \left(\mathbf{y}_t - \mathbf{H}\mathbf{x}_t^f\right)\right), \tag{2}$$

where $\mathbf{\Sigma}_t = \mathbf{H}\mathbf{P}_t^f\mathbf{H}^\top + \mathbf{R}$, with $\mathbf{P}_t^f = \mathbf{M}\mathbf{P}_{t-1}^a\mathbf{M}^\top + \mathbf{Q}$ and $\mathbf{P}_{t-1}^a$ corresponds to the state covariance estimated by the Kalman filter at time $t-1$. The innovation likelihood given in Eq. (2) is interesting because it corresponds to the squared distance between the observations and the forecast normalized by their uncertainties, represented by the covariance $\mathbf{\Sigma}_t$.

At each iteration of the augmented Kalman procedure, the estimate of the matrix $\mathbf{M}$ is given by the least square estimator, using a linear regression such that:

$$\mathbf{M}^{(i)} = \sum_{t=2}^{T} \frac{\left(\mathbf{x}_{t-1}^{(i-1)}(\mathbf{x}_{t-1}^{(i-1)})^\top\right)^{-1}\mathbf{x}_t^{(i-1)}(\mathbf{x}_{t-1}^{(i-1)})^\top}{T-1}, \tag{3}$$

where $\mathbf{x}^{(i-1)}$ corresponds to the output catalog of the previous iteration (result of a Kalman smoothing and a Gaussian sampling, explained more in details below). Following Eq. (1a), the covariance $\mathbf{Q}$ is estimated empirically using the estimate of $\mathbf{M}$ given in Eq. (3), such that:

$$\mathbf{Q}^{(i)} = \sum_{t=2}^{T} \frac{\left(\mathbf{x}_t^{(i-1)} - \mathbf{M}^{(i)}\mathbf{x}_{t-1}^{(i-1)}\right)\left(\mathbf{x}_t^{(i-1)} - \mathbf{M}^{(i)}\mathbf{x}_{t-1}^{(i-1)}\right)^\top}{T-1}. \tag{4}$$

Then, a Kalman smoother is applied using the $\mathbf{M}^{(i)}$ and $\mathbf{Q}^{(i)}$ matrices estimated in Eq. (3) and Eq. (4). At each time $t$, it results to a Gaussian mean vector $\mathbf{x}_t^s$ and a covariance matrix $\mathbf{P}_t^s$. As input of the next iteration of the algorithm, the catalog $\mathbf{x}^{(i)}$ is updated using a Gaussian random sampling using $\mathbf{x}_t^s$ and $\mathbf{P}_t^s$ at each time $t$. This random sampling is used to exploit the linear correlations between the components of the state vector, that appear in the non-diagonal terms of $\mathbf{P}^s$. The random sampling is also used to avoid being trapped in a local maximum, as in stochastic EM procedures (Delyon et al., 1999).

The likelihood calculated at each iteration of the procedure increases until convergence. The algorithm is stopped when the likelihood difference between two iterations becomes small. The solution of the proposed method is the last Gaussian mean vectors $\mathbf{x}_t^s$ and covariance matrices $\mathbf{P}_t^s$ calculated at each time $t$. The component corresponding to the latent component $\mathbf{z}$ is finally retrieved, with an information about its uncertainty.

## 3 Experiment and evaluation metrics

The methodology is tested on the Lorenz-63 system (Lorenz, 1963). This 3-dimensional dynamical system models the evolution of the convection ($x_1$) as a function of horizontal ($x_2$) and vertical temperature gradients ($x_3$). The evolution of the system is governed by three ordinary differential equations such as:

$$\dot{x_1} = 10(x_2 - x_1), \tag{5a}$$

$$\dot{x_2} = x_1(28 - x_3) - x_2, \tag{5b}$$

$$\dot{x_3} = x_1 x_2 - \frac{8}{3}x_3. \tag{5c}$$

Runge-Kutta 4-5 is used to integrate the Lorenz-63 equations to generate x1, x2, and x3. In this paper, it is assumed that $x_1$ is never observed, only $x_2$ and $x_3$ are observed on 10 model time units of the Lorenz-63 system, every $dt = 0.001$ time steps (top of Fig. 2). The observation vector is thus $\mathbf{y} = [y_2, y_3]$. In what follows, only those data are available, not the set of Eqs. (5).

The methodology is applied to the Lorenz-63 system, adding sequentially a new hidden component in the state of the system as follow. At the beginning, the state is augmented such that $\mathbf{x} = [x_2, x_3, z_1]$, where $z_1$ is randomly initialized with a white noise, with variance $\sigma^2 = 5$. The observations are stored in the vector $\mathbf{y} = [y_2, y_3]$. The observation operator is thus the $2 \times 3$ matrix $\mathbf{H} = [1, 0, 0|0, 1, 0]$. After 30 iterations of the algorithm presented in section 2, the hidden component $z_1$ has converged. After that, a new white noise $z_2$ is used to augment the state such that $\mathbf{x} = [x_2, x_3, z_1, z_2]$, the vector $\mathbf{y} = [y_2, y_3]$ remains the same, and the iterative algorithm is applied until stabilization of $z_2$. As long as the stabilized likelihood continues to increase with the addition of a hidden component, this state augmentation procedure is repeated.

Note that several hidden components can be added all at once, with similar performance as the sequential procedure described above (results not shown). In this all at once case, the interpretation of the retrieved components is not as informative, thus we decided to retain the sequential case. Note also that the methodology has been tested with larger $dt$ (i.e., 0.01 and 0.1). The conclusion is that by increasing the time delay between observations, it significantly increases the number of latent variables (results not shown). Finally, the assimilation window length corresponds to $10^4$ time steps. By reducing this length (e.g., to $10^3$, $10^2$, $10^1$), the conclusions remain the same as for $dt = 0.001$.

## 4 Results

Using the experiment presented in section 3, three hidden components $z_1$, $z_2$, and $z_3$ were sequentially added. They are reported in Fig. 2, as well as the true Lorenz components $x_1$, $x_2$, and $x_3$. Although they do not fit the hidden variable $x_1$ of the Lorenz-system, the two first hidden components $z_1$ and $z_2$ show time variations. On the contrary, $z_3$ remains close to 0, with a large confidence interval. This suggests that our method has identified that 2 hidden variables are enough to retrieve the dynamics of the 2 observed variables. This result is consistent with the effective dimension of the Lorenz-63 system, which is between two and three. Here, as the estimated dynamical model $\mathbf{M}$ is a linear approximation, the dimension of the augmented state and the observed components is higher than the effective one.

This is confirmed by the evaluation of the likelihood of the observations $y_2$ and $y_3$ with different linear models, obtained with or without the use of hidden components $\mathbf{z}$ (Fig. 3). This likelihood is useful to diagnose the optimal number of dimensions needed to emulate the dynamics of the observed components. As the proposed method is stochastic, 50 independent realizations of the likelihood are shown for each experiment. The 50 realizations vary from the random values given to the added hidden variable at the beginning of the iterative procedure. In the naive case where the state of the system is $[x_2, x_3]$ (black dashed line), the likelihood is small. Then, adding successively $z_1$ (green lines) and $z_2$ (red lines), after 30 iterations of the proposed algorithm, the likelihood significantly increases. Finally, due to a significant increase of the forecast covariance $\mathbf{P}^f$ in Eq. (2), the inclusion of $z_3$ reduces the likelihood (purple lines). This suggests that a third variable is not needed, and is even detrimental to the skill of the reconstruction. Those results indicate that the best linear model to predict the variations of the observations

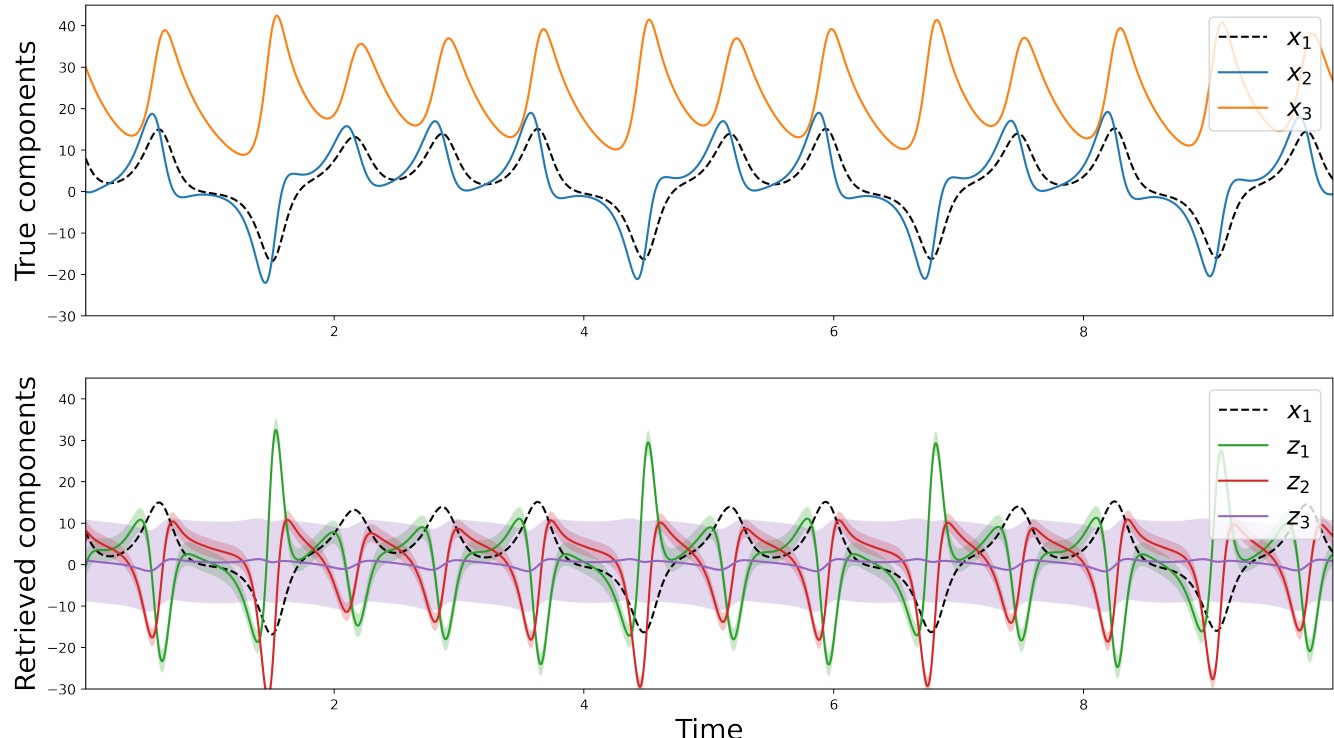

**Figure 2.** True components of the Lorenz-63 model (top) and hidden components estimated using the iterative and augmented Kalman procedure (bottom). The shaded colors corresponds to the 95% Gaussian confidence intervals.

$y_2$ and $y_3$ is the one using two hidden components. Thus, for the rest of the paper, the focus is thus done on the model with the following augmented state $\mathbf{x} = [x_2, x_3, z_1, z_2]$.

The question is now: what is the significance of those hidden components $z_1$ and $z_2$ estimated using the proposed methodology? Are they correlated to the unobserved component $x_1$ or to the observed one $x_2$ and $x_3$? Are they somehow proxies of the unobserved component? Using symbolic regression (i.e., using basic mathematical transformations of $x_2$ and $x_3$ as regressors to explain $z_1$ and $z_2$), it has been found that the hidden components $\mathbf{z}$ correspond to linear combinations of the derivatives of the observations such that:

$$z_1 = a_2\dot{x}_2 + a_3\dot{x}_3, \tag{6a}$$

$$z_2 = b_1\dot{z}_1 + b_2\dot{x}_2 + b_3\dot{x}_3. \tag{6b}$$

When developing Eq. (6b) using Eq. (6a), the second hidden component writes $z_2 = b_2\dot{x}_2 + b_3\dot{x}_3 + b_1a_2\ddot{x}_2 + b_1a_3\ddot{x}_3$. It shows that $z_1$ uses the first derivative of $x_2$ and $x_3$, whereas $z_2$ uses the second derivatives. This result makes the link with Taylor's and Takens' theorem, which shows that an unobserved component (i.e., $x_1$), can be replaced by the observed components (i.e., $x_2$ and $x_3$) at different time lags. Note that due to the stochastic behaviour of the algorithm, the $a$ and $b$ coefficients

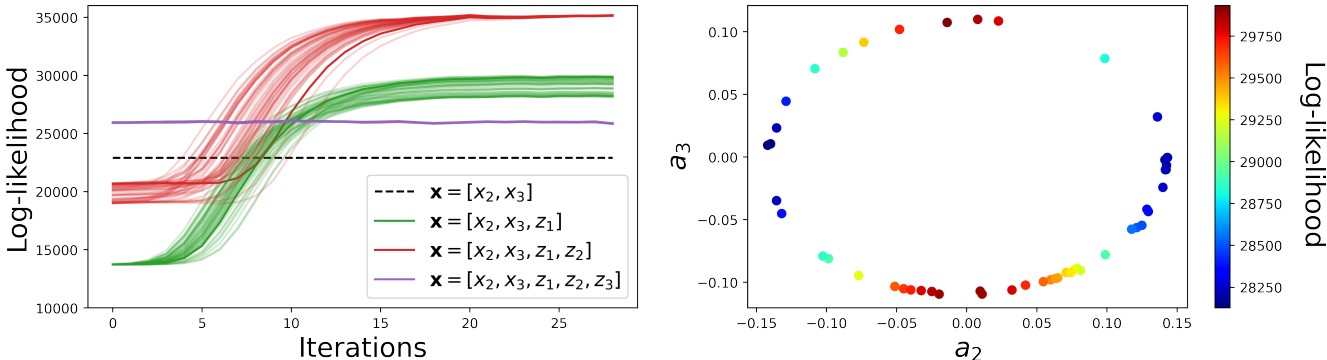

**Figure 3.** Likelihoods as a function of the iteration of the augmented Kalman procedure (left) and estimation of the $a_2$ and $a_3$ parameters (right). Different dynamical models are considered, from none to three hidden components in $\mathbf{z}$, whereas only $x_2$ and $x_3$ are observed in the Lorenz-63 model. The likelihood of 50 independent realizations of the iterative and augmented Kalman procedure are shown.

are not fixed and several combination of them can reach to the same performance in term of likelihood. This is illustrated in Fig. 3 (left panel), with 50 independent realizations of the proposed algorithm. When considering only $z_1$ (green lines), the
algorithm converges to various solutions, mainly restricted around two solutions (corresponding to a minimum and a maximum of likelihood). As shown in Fig. 3 (right panel), the minimum likelihood corresponds to $a_3 = 0$ and the maximum likelihood corresponds to $a_2 = 0$. Thus, the likelihood when $z_1 = a_3 \dot{x}_3$ is higher than when $z_1 = a_2 \dot{x}_2$. This suggests that $\dot{x}_3$ is more important than $\dot{x}_2$ to explain the variations of the Lorenz system (this is consistent with investigation of Sévellec and Fedorov, 2014, in a modified version of Lorenz-63 model). Interestingly, the scatter plot between $a_2$ and $a_3$ shows a circular relationship.
This is also the case for $b_2$ and $b_3$ (results not shown). Then, in Fig. 3 (left panel), when considering $z_1$ and $z_2$ (red lines), the 50 independent realizations reach the same likelihood after 30 iterations. It means that if $a_3 = 0$ when considering only $z_1$, then $b_3 \neq 0$ when introducing $z_2$. In terms of forecast performance, this is similar to $a_2 = 0$ and $b_2 \neq 0$, because the likelihoods converge to the same value (red lines after 30 iterations).

To compare the performance of the naive linear model $\mathbf{M}$ with $[x_2, x_3]$ and the ones with $[x_2, x_3, z_1]$ or $[x_2, x_3, z_1, z_2]$, their
forecasts are evaluated. After applying the proposed algorithm, the $\widehat{\mathbf{M}}$ and $\widehat{\mathbf{Q}}$ estimated matrices are used to derive probabilistic forecast, starting from the last available observation $\mathbf{y}_t$, using:

$$\mathrm{E}[\mathbf{x}_{t+1}|\mathbf{y}_1,\ldots,\mathbf{y}_t] = \widehat{\mathbf{M}}\mathrm{E}[\mathbf{x}_t|\mathbf{y}_1,\ldots,\mathbf{y}_t], \tag{7a}$$

$$\mathrm{Cov}[\mathbf{x}_{t+1}|\mathbf{y}_1,\ldots,\mathbf{y}_t] = \widehat{\mathbf{M}}\mathrm{Cov}[\mathbf{x}_t|\mathbf{y}_1,\ldots,\mathbf{y}_t]\widehat{\mathbf{M}}^T + \widehat{\mathbf{Q}}, \tag{7b}$$

with E and Cov, the expectation and the covariance, respectively. To test the predictability of the different linear models
(i.e., with or without hidden components $\mathbf{z}$), a test set has been created, starting from the end of the sequence of observations $(\mathbf{y}_1,\ldots,\mathbf{y}_T)$ used in the assimilation window. This test set is also corresponding to $10^4$ time steps with $dt = 0.001$. It is used to compute two metrics, the Root Mean Square Error (RMSE) and the coverage probability at $50\%$. The RMSE is used to evaluate the precision of the forecasts, comparing the true $x_2$ and $x_3$ components to the estimated ones, whereas the coverage

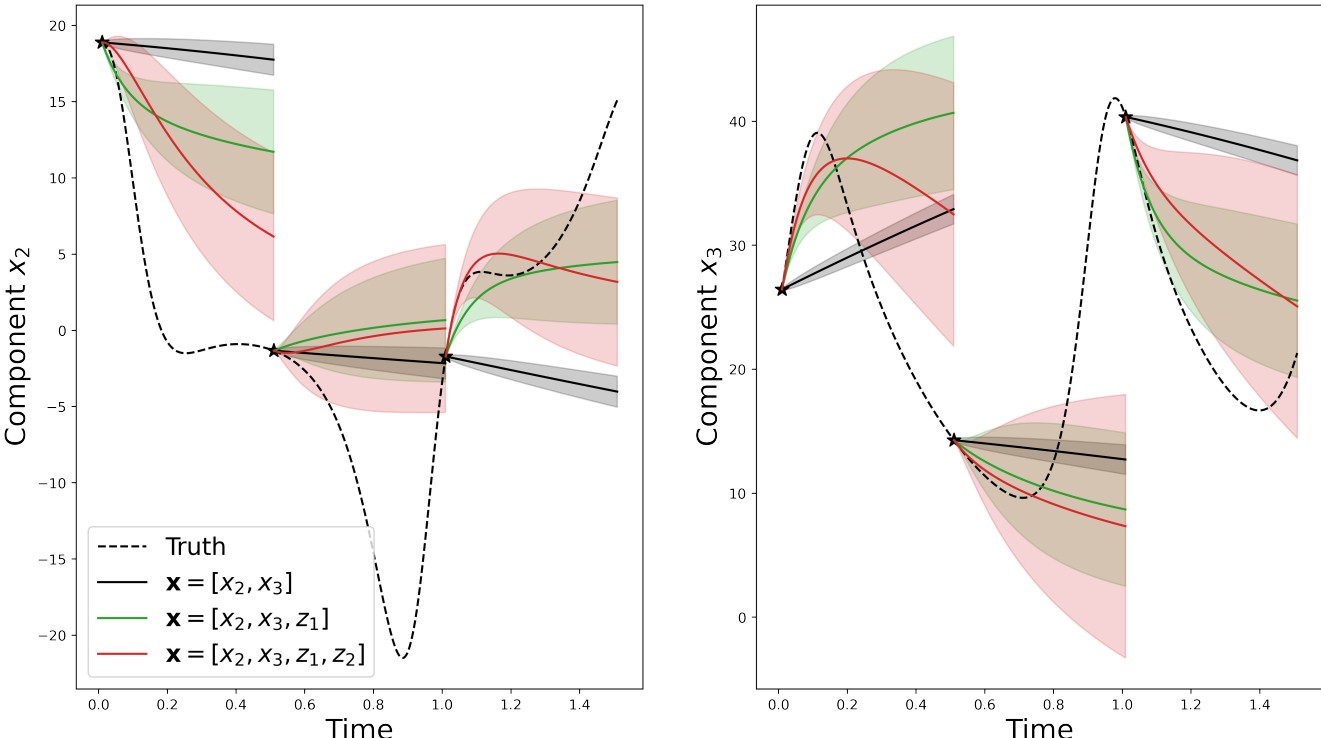

**Figure 4.** Example of three statistical forecasts of $x_2$ (left) and $x_3$ (left) with their $50\%$ prediction interval using 3 different linear operators with: no hidden component (dashed black), one hidden component (green), and two hidden components (red). These predictions are obtained using sequential statistical forecasts, as explained in Eqs. (7), on an independent test dataset.

probability is used to evaluate the reliability of the prediction, evaluating the proportion of true trajectories falling within the
$50\%$ prediction interval of $x_2$ and $x_3$. Examples of predictions are given in Fig. 4. It shows bad linear predictions of the model
with only $[x_2, x_3]$ (dashed black lines). As the $\mathbf{M}$ operator is not time-dependent, the predictions are quite similar, close to the
persistence. Then, adding one (green) or two (red) hidden components in the $\mathbf{M}$ operators creates some nonlinearities in the
forecasts.

In Fig. 5, the predictions are evaluated over the whole test dataset, for different lead times. By introducing hidden compo-
nents, the RMSE decreases for both $x_2$ and $x_3$ components (top panels). For instance, for a lead time of $0.05$, when considering
two hidden components, the RMSE is halved when it is compared to the naive linear model without hidden components. The
coverage probability metric is also largely improved (bottom panels). Indeed, the results with two hidden components are close
to $50\%$, the optimal value.

To evaluate where the linear model with $[x_2, x_3, z_1, z_2]$ performs better than the one with $[x_2, x_3]$, the Euclidean distances
between the forecasts (for a lead time of $0.1$) and the truth are computed. Those errors are evaluated at each time step of the test
dataset, in the $(x_2, x_3)$ space. Based on those errors, Fig. 6 shows the relative improvement between the model without and the

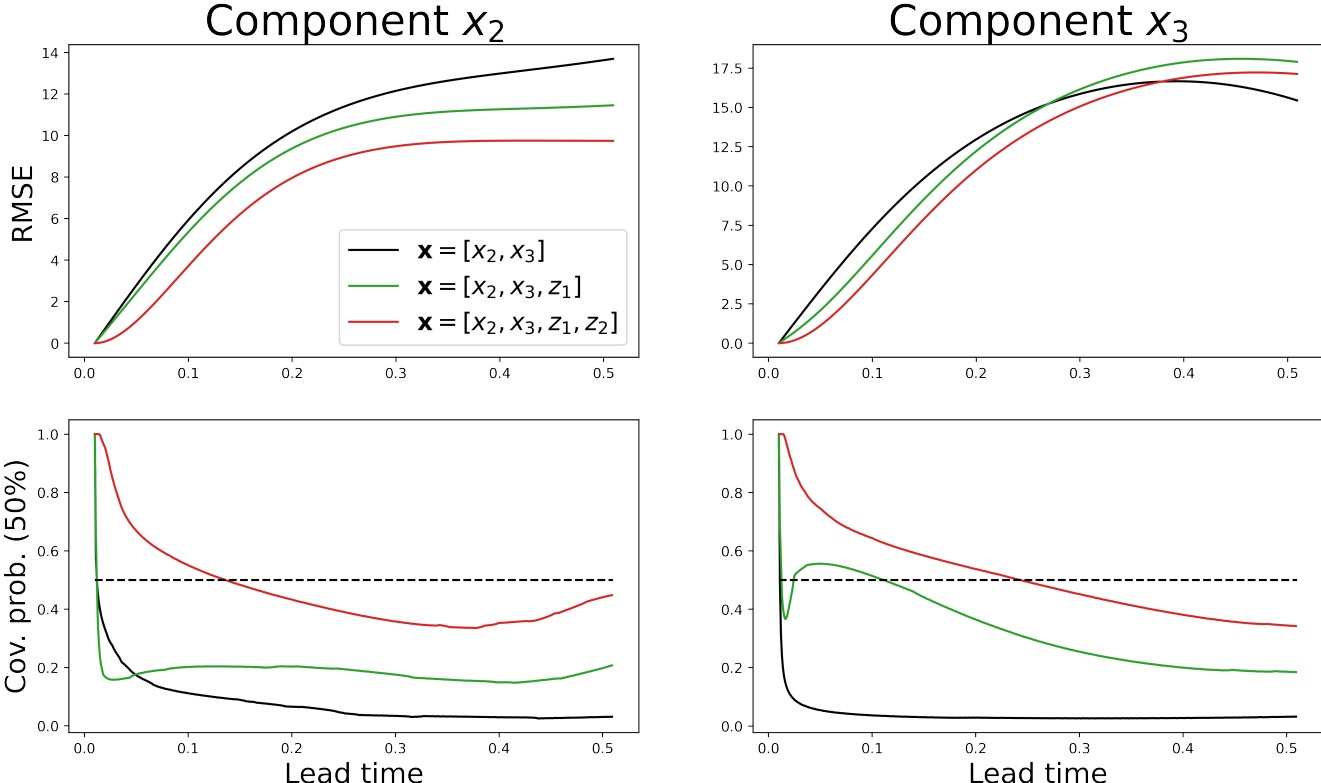

**Figure 5.** Root Mean Square Error (top) and 50% coverage probability (bottom) as a function of the lead time (x-axis) for the reconstruction of the components $x_2$ (left) and $x_3$ (right). These metrics are evaluated on an independent test dataset.

model with hidden components. When the two models have similar performance, values are close to $0$ (white), and when the model including $z_1$ and $z_2$ is better, values are close to $1$ (red). Figure 6 clearly shows that error reduction is not homogeneous in the attractor. The improvement is moderate in the outside of the wings of the attractor, but important in the wing-transition. It suggests that the introduction of the hidden components $z_1$ and $z_2$ makes it possible to provide information on the position in the attractor and thus to make better predictions, especially in bifurcation regions.

## 5 Conclusions

In this article, the goal is to retrieve hidden components of a dynamical system that is partially observed. The proposed methodology is purely data-driven, not physics-driven (i.e., without the use of any equations of the dynamical model). It is based on the combination of data assimilation and machine learning techniques. Three main ideas are used in the methodology: an augmented state strategy, a linear approximation of a dynamical system, and an iterative procedure. The methodology is

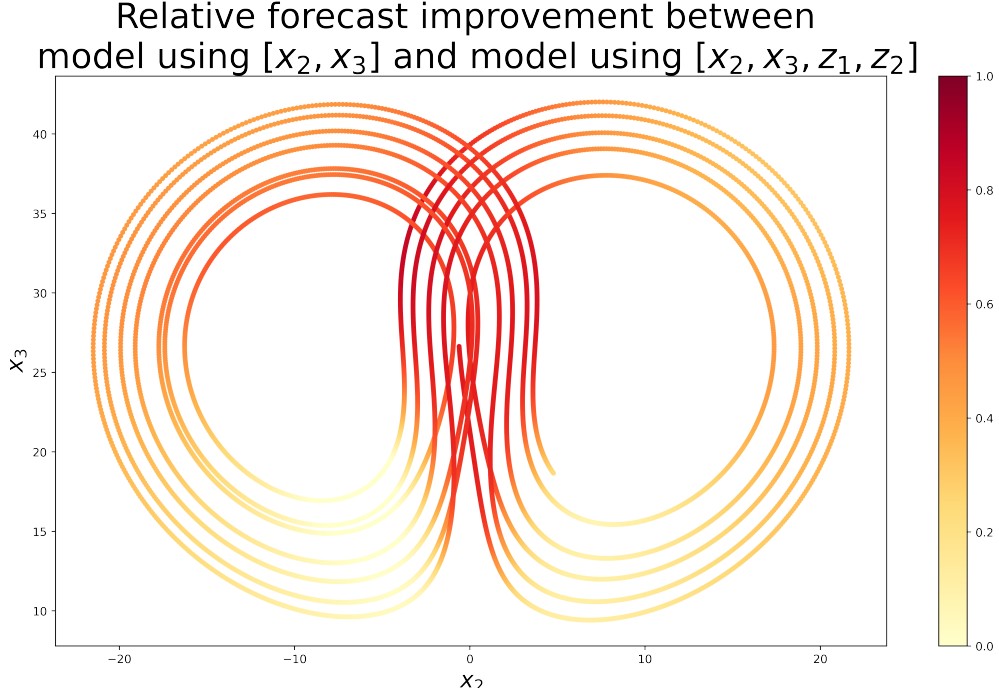

**Figure 6.** Relative forecast improvement measured as 1 minus the ratio between two Euclidean distances: the one calculated with model $[x_2, x_3, z_1, z_2]$ (at numerator) and the one calculated with model $[x_2, x_3]$ (at denominator). The Euclidean distances are calculated in the $(x_2, x_3)$ space and correspond to the error between the forecasts (for a lead time of $0.1$) and the truth, evaluated on an independent test dataset.

easy to implement, using simple strategies and well established algorithms: Kalman filter and smoother, linear regression using least squares, iterative procedure inspired from the EM recursions, and Gaussian random sampling for the stochastic aspect.

The methodology is tested on the Lorenz-63 system, where only two components of the system are observed on a short period of time. Several hidden components are introduced sequentially in the system. Although the hidden components are initialized randomly, only a few iterations of the proposed algorithm is necessary to retrieve a relevant information. The recovered components are expressed with Gaussian distributions. The new components correspond to linear combinations of successive derivatives of the observed variables. This result is consistent with the theorems of Taylor and Takens which show that time delay embedding is useful to improve the forecasts of the system. In our case, this is evaluated using the likelihood: a metric to evaluate the innovation (i.e., the difference between Gaussian forecasts and Gaussian observations).

Using our methodology, we do not retrieve the true missing Lorenz component and need two hidden variables to represent a single missing one. The reason of this mismatch is two-fold and is mainly due to the linear approximation of the dynamical system, which implies that: (1) the true missing component, that does not have to be linear combinations of the observed variables, is impossible to retrieve in our framework and (2) two variables, using combinations of the time derivatives of the

observed variables, are needed to accurately represent the complexity of the dynamics. However, it is important to note that, even if two variables are needed to replace a single one, the dynamical evolution of the system is relatively well captured, for short lead times, with our methodology. This correct representation of the evolution might ultimately be the most important (e.g., for accurate and reliable forecasting).

The proposed methodology is using a strong assumption: the linear approximation of the dynamical system is global (i.e., fixed for the whole observation period). A perspective is to use adaptive approximations of the model using local linear regressions. This strategy is computationally more expensive because a linear regression is adjusted at each time step, but shows some improvements in chaotic systems (see Platzer et al., 2021a, b). In this context of adaptive linear dynamical model, the proposed methodology could be easily plugged into an ensemble Kalman procedure based on analog forecasts (Lguensat et al., 2017). In futur works, we plan to compare the global and local linear approaches (i.e., fix or adaptive linear surrogate model). We also plan to compare them to nonlinear surrogate models, based on neural network architectures with latent information encoded in an augmented space or in hidden layers (e.g., LSTM).

In this paper, we have demonstrated the feasibility of the method on an idealized and comprehensive problem, using the Lorenz-63 system. In the future, we plan to apply the methodology to more challenging problems, like the Lorenz-96 system or a quasi-geostrophic model. For the application on real data, we plan to use a database of observed climate indices and try to find latent variables that help to make data-driven predictions.

*Author contributions.* Pierre Tandeo wrote the article. Pierre Tandeo and Pierre Ailliot developed the algorithm. Florian Sévellec and Pierre Ailliot helped on the redaction of the paper.

*Competing interests.* No competing interests are present.

*Acknowledgements.* This paper is the result of a project proposed in a course on "Data Assimilation" in the master program "Ocean Data Science" at Univ. Brest, ENSTA Bretagne, and IMT Atlantique, France. Authors would like to thanks the students for their implications in the project: Nils Niebaum, Zackary Vanche, Benoit Presse, Dimitri Vlahopoulos, Yanis Grit, and Joséphine Schmutz. The authors would like to thank Noémie Le Carrer for her proofreading of the paper, as well as Paul Platzer, Said Ouala, Lucas Drumetz, Juan Ruiz, Manuel Pulido, and Takemasa Miyoshi for their valuable comments. This work was supported by ISblue project, Interdisciplinary graduate school for the blue planet (ANR-17-EURE-0015) and co-funded by a grant from the French government under the program "Investissements d'Avenir" embedded in France 2030. This work was also supported by LEFE program (LEFE IMAGO projects ARVOR).

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
