# Peer review of "Data-driven Reconstruction of Partially Observed Dynamical Systems"

_EGUsphere, 2022_

## Referee Comment (RC1)

**Data-driven Reconstruction of Partially Observed Dynamical Systems – review report**

16th January 2023

In this manuscript, the authors derive a method to reconstruct the dynamics of a system from partial observations, in which data assimilation and machine learning steps are alternate. The data assimilation steps are used to estimate the state from observations using the surrogate model, while the machine learning steps are used to estimate the surrogate model from the data assimilation analysis. This method is the same as the one derived by Brajard et al. 2020, with the exception that, on top of this method, the authors propose a new, innovative state augmentation process. The entire method is illustrated using numerical experiments with the 3-variable Lorenz 1963 system.

I am overall positive about this manuscript. The text reads very well and is easy to follow. To my knowledge, the state augmentation process is new and deserves to be published. However, I have some concerns, in particular about the methodology and about the experiments, that needs to be fixed before I can recommend publication.

**1 General comments**

**1.1 How the methodology differs from that of Brajard et al. (2021)**

As far as I understand, the method derived in this manuscript proposes to alternate data assimilation steps (with the ensemble Kalman smoother) and machine learning steps (with a linear regression) on a given dataset of observations until convergence. This is exactly what has been originally proposed by Brajard et al. (2020) and later formalised by Bocquet et al. (2020). Pushing further the comparison, I see only three significant differences with the original method:

- in the present method the machine learning step is restricted to linear regression, while in the original method, nonlinear regression tools (such as neural networks) are used;

- in the present method observations are assumed to be perfect (even though they are sparse), while in the original method, sparse and noisy observations are used;

- the state augmentation process added on top the data assimilation / machine learning iterations.

I do not see the first two points as a major limitations, in fact I am rather confident that the present method should also work with neural networks replacing the linear regression and with noisy observations. By contrast, the third point is in my opinion the real added value of the present work, and this should be emphasised.

**Additional questions about the methodology**

1. Is there a fundamental reason to use only linear regression and perfect observations? If not, I would suggest to get rid of these assumptions in the methodological section.

2. How does the state augmentation scale with the system dimension?

3. Can the additional state components be added all at once? Did you try that in the numerical experiments?

4. In the experiments, 30 iterations seem sufficient to reach convergence. Do you have an idea how this number would scale with the system dimension?

5. The text is ambiguous about the data assimilation method used: 'and thus uses the classic Kalman filter and smoother equations' (L 71-72), 'by the Kalman filter' (77-78) 'a Kalman smoother is applied' (L 87) 'Kalman filter and smoother' (L 161). Kalman filter or smoother, you have to choose (I assume it is Kalman smoother).

**1.2 About the numerical experiments**

The description of the experiments is incomplete, in such a way that the experiments cannot be reproduced without further assumptions. For example, what numerical method is used to integrate in time the model equations to compute the truth?

Furthermore, I have a serious concern about the 'model distance' introduced by equation (6). Without further details, I assume that it is computed using the same trajectory as the training step. Using the same data for training and testing should be avoided by all means. Moreover, in this context where observation are perfect, I am not sure to see the point of this metric: observations are required to initialise the model (for the hidden components), but if we have observations, we do not need the forecasting system any more since observations are perfect... Therefore, I think that the metric used to evaluate the accuracy of the model should be reconsidered.

**Additional questions about the experiments**

1. '10 loops of the Lorenz-63 system' (L 104-105) Do you mean 10 model time units or 10 revolutions on the model attractor? In any case, I would not say that this is a small period of time, compared to the doubling time which is 0.78 MTU.

2. From what I understand (L 104-106), you have access to the true $x_2$ and $x_3$ (no observation noise) every $dt = 0.001$ (which is probably the integration time step for the truth). This seems to be very strong requirements. Can you discuss this?

3. What is the choice of the data assimilation window length for the ensemble Kalman smoother? Without further details, I assume that it covers the entire experiment, i.e. $10^4$ observation steps. This is really huge. Can you discuss this?

**2 Technical comments and suggestions**

**L 17-18**  'using Bayesian framework' → 'using a Bayesian framework'?

**L 21**  'All the approaches cited above are assuming that the full state of the system is observed' This is not true: at least Tandeo et al. (2015), Lguensat et al. (2017), Bocquet et al. (2019), Brajard et al. (2020), Fablet et al. (2021) use sparse observation operators in their methods. I would replace 'All the approaches cited above' by 'Many approaches'.

**L 23-24**  'To deal with those strong constraints' I would replace here 'constraints' by 'assumptions' in order to avoid a potential confusion with strong-constraint methods in variational data assimilation.

**L 24-26**  'An option is to [...] whereas an other option is to [...]' I would suggest to also mention here the combination of data assimilation and machine learning, because (i) this is what is used in some of the previously cited papers (the ones that can handle sparse and noisy observations), and (ii) this is what is used in the present manuscript!

**L 29**  'with a dynamical model (model- or data-driven)' I would replace here 'model-driven' by 'based on physical knowledge' or something like this (to avoid a *model-driven model*).

**L 31**  'estimation of the parameters' Which parameters?

**L 40-41**  'from data assimilation, machine learning, and theory of dynamical systems' → 'from data assimilation, machine learning, and dynamical systems'?

**L 42**  'from partial observations $\mathbf{y}$' In data assimilation, observations are usually noisy in addition to being partial.

**L 42-46**  In this paragraph, why didn't you mention the crucial role of the background error statistics?

**L 48**  'to mathematically approximate the dynamic of the system' → 'to mathematically approximate the system dynamics'.

**L 76**   In equation (2), I would suggest to explicit the definition of $\mathcal{L}$, i.e. use something like that:

$$\mathcal{L} \triangleq p(\mathbf{y}_1, \ldots \mathbf{y}_T | \mathbf{x}_1^f, \ldots \mathbf{y}_T^f) \propto \prod_{t=1}^{T} \cdots \tag{1}$$

Furthermore, $T$ is undefined in this equation.

**L 78-79**   'The innovation likelihood given in Eq. (2) is interesting because it corresponds to the squared distance between the observations and the forecast normalized by their uncertainties, represented by the covariance $\Sigma_t$.' In data assimilation, this quantity is simply called 'the likelihood'.

**L 89-90**   'This random sampling is used to exploit the correlations between the components of the state vector' I do not understand why this is necessary. Could you elaborate?

**L 110**   'After 30 iterations of the algorithm presented in section 2, the hidden component $z_1$ is stabilized.' Can you please explain the exact meaning of 'stabilized' in this context?

**L 114**   'this augmented state procedure is repeated' → 'this state augmentation process is repeated'.

**L 117**   '$z_3$ is very flat' I would replace 'very' by 'rather' in this statement.

**L 125**   'Finally, the inclusion of $z_3$ reduces the likelihood (purple lines).' Do you have an explanation for this phenomenon?

**L 131**   In equation (6), I would explicit the dependence on time, i.e. replace dist($\mathbf{M}$) by dist($\mathbf{M}$)($t$).

**L 137-138**   'Are they correlated with the unobserved component $x_1$ or with the observed one $x_2$ and $x_3$?' → 'Are they correlated to the unobserved component $x_1$ or to the observed ones $x_2$ and $x_3$?'?

**L 139**   'It has been found that...' How did you come up with this? As it is presented, it looks like something pulled out of a hat.

**L 47-48**   'This is illustrated in Fig. (3), with 50 independent realizations of the proposed algorithm.' Strictly speaking, this is not the case since $a$ and $b$ are not represented in this figure.

**L 152-153** 'Then, when considering $z_1$ and $z_2$ (red lines), the 50 independent realizations reach the same likelihood after 30 iterations.' What about $a$ and $b$? Are they similar over the 50 realisations?

**L 153-154** 'it will then focuses' → 'it will then focus'.

**L 175-176** 'the dynamical evolution of the system is retrieved with our methodology' This is not clearly shown in the experiments.

**References**

Bocquet, Marc, Julien Brajard, Alberto Carrassi and Laurent Bertino (2020). 'Bayesian inference of chaotic dynamics by merging data assimilation, machine learning and expectation-maximization'. In: *Foundations of Data Science* 2.1, pp. 55–80. DOI: 10.3934/fods.2020004.

---

## Author Comment (AC1)

**Response to Reviewer 1**

Data-Driven Reconstruction of Partially Observed Dynamical Systems, by Tandeo et al.

**Note: your comments and questions are reported in this document and we use bold text for our responses.**

In this manuscript, the authors derive a method to reconstruct the dynamics of a system from partial observations, in which data assimilation and machine learning steps are alternate. The data assimilation steps are used to estimate the state from observations using the surrogate model, while the machine learning steps are used to estimate the surrogate model from the data assimilation analysis. This method is the same as the one derived by Brajard et al. 2020, with the exception that, on top of this method, the authors propose a new, innovative state augmentation process. The entire method is illustrated using numerical experiments with the 3-variable Lorenz 1963 system. I am overall positive about this manuscript. The text reads very well and is easy to follow. To my knowledge, the state augmentation process is new and deserves to be published. However, I have some concerns, in particular about the methodology and about the experiments, that needs to be fixed before I can recommend publication.

**Thank you for your encouraging review. Below you will find the responses to your different points.**

1 General comments

1.1 How the methodology differs from that of Brajard et al. (2021)

As far as I understand, the method derived in this manuscript proposes to alternate data assimilation steps (with the ensemble Kalman smoother) and machine learning steps (with a linear regression) on a given dataset of observations until convergence. This is exactly what has been originally proposed by Brajard et al. (2020) and later formalised by Bocquet et al. (2020). Pushing further the comparison, I see only three significant differences with the original method:
• in the present method the machine learning step is restricted to linear regression, while in the original method, nonlinear regression tools (such as neural networks) are used;
• in the present method observations are assumed to be perfect (even though they are sparse), while in the original method, sparse and noisy observations are used;
• the state augmentation process added on top the data assimilation / machine learning iterations.
I do not see the first two points as a major limitations, in fact I am rather confident that the present method should also work with neural networks replacing the linear regression and with noisy observations. By contrast, the third point is in my opinion the real added value of the present work, and this should be emphasised.

**Indeed, the added value is the state augmentation. This is the core of the paper, and, we feel, the innovative part. We also wanted to remind that the alternance of DA (Kalman linear) and ML (linear regression) is not new and has been proposed in the context of the Expectation Maximization (EM) algorithm. We have clarified this point in the new version of the manuscript, l. 47: "The current paper is thus an extension of (Shumway and Stoffer, 1982) to never observed components of a dynamical system, using a state augmentation strategy."**

**Additionally, we now mention l. 33 the following paper, which was not cited previously: Brajard et al. 2021.**

Additional questions about the methodology

1. Is there a fundamental reason to use only linear regression and perfect observations? If not, I would suggest to get rid of these assumptions in the methodological section.

**Indeed, there is no reason to only consider this simple case. This is now stated in l. 77: "Nonlinear and adaptive operators as well as noisy observations could be taken into account but, for the sake of simplicity, only the linear and non-noisy case is considered in this paper."**

2. How does the state augmentation scale with the system dimension?

**Thank you for this interesting question. We added this discussion l. 153: "This result is consistent with the effective dimension of the Lorenz-63 system, which is between two and three. Here, as the estimated dynamical model M is a linear approximation, the dimension of the augmented state and the observed components is higher than the effective one." We also point out the role of the likelihood to find the number of hidden components, l. 157: "This likelihood is useful to diagnose the optimal number of dimensions needed to emulate the dynamics of the observed components."**

3. Can the additional state components be added all at once? Did you try that in the numerical experiments?

**Yes, indeed it is possible to add all the latent variables at the same time. This is now clarified in l. 142: "Note that several hidden components can be added all at once, with similar performance as the sequential procedure described above (results not shown). In this all at once case, the interpretation of the retrieved components is not as informative, thus we decided to retain the sequential case." We decided to retain the iterative strategy, especially to introduce and explain clearly Eqs. (6a) and (6b).**

4. In the experiments, 30 iterations seem sufficient to reach convergence. Do you have an idea how this number would scale with the system dimension?

**There is no clear relationship between the number of EM iterations and the dimension of the system, and this point doesn't seem to be discussed in the literature. However, the EM algorithm has a slow (linear) asymptotic convergence speed, but is generally**

efficient in the first iterations to quickly increase the likelihood, and provide a first estimate of the parameters.

5. The text is ambiguous about the data assimilation method used: 'and thus uses the classic Kalman filter and smoother equations' (L 71-72), 'by the Kalman filter' (77-78) 'a Kalman smoother is applied' (L 87) 'Kalman filter and smoother' (L 161). Kalman filter or smoother, you have to choose (I assume it is Kalman smoother).

**The two Kalman recursions are necessary in this paper and we decided to keep this distinction because, as now stated in l. 98: "The Kalman filter (forward in time) is used to get the information of the likelihood, whereas the Kalman smoother (forward and backward in time) is used to get the best estimate of the state."**

1.2 About the numerical experiments
The description of the experiments is incomplete, in such a way that the experiments cannot be reproduced without further assumptions. For example, what numerical method is used to integrate in time the model equations to compute the truth? Furthermore, I have a serious concern about the 'model distance' introduced by equation (6). Without further details, I assume that it is computed using the same trajectory as the training step. Using the same data for training and testing should be avoided by all means. Moreover, in this context where observation are perfect, I am not sure to see the point of this metric: observations are required to initialise the model (for the hidden components), but if we have observations, we do not need the forecasting system any more since observations are perfect... Therefore, I think that the metric used to evaluate the accuracy of the model should be reconsidered.

**Thank you for those remarks about the numerical experiments. It is now stated in l. 132: "Runge-Kutta 4-5 is used to integrate the Lorenz-63 equations to generate x1, x2, and x3."**

**Regarding the Eq. (6) and the metric of evaluation, it has been completely modified, taking into account the remarks of the two Reviewers. It now reads, l. 189: "To compare the performance of the naive linear model M with [x2, x3] and the ones with [x2, x3, z1] or [x2, x3, z1, z2], their forecasts are evaluated. After applying the proposed algorithm, the M and Q estimated matrices are used to derive probabilistic forecast, starting from the last available observation y_t, using:**

$$E[\mathbf{x}_{t+1}|\mathbf{y}_1,\ldots,\mathbf{y}_t] = \widehat{\mathbf{M}}E[\mathbf{x}_t|\mathbf{y}_1,\ldots,\mathbf{y}_t],$$

$$Cov[\mathbf{x}_{t+1}|\mathbf{y}_1,\ldots,\mathbf{y}_t] = \widehat{\mathbf{M}}Cov[\mathbf{x}_t|\mathbf{y}_1,\ldots,\mathbf{y}_t]\widehat{\mathbf{M}}^T + \widehat{\mathbf{Q}},$$

**with E and Cov, the expectation and the covariance, respectively. To test the predictability of the different linear models (i.e., with or without hidden components z), a test set has been created, starting from the end of the sequence of observations (y_1, ..., y_T) used in the assimilation window. This test set is also corresponding to 10^4 time steps with dt=0.001. It is used to compute two metrics, the Root Mean Square Error (RMSE) and the coverage probability at 50%. The RMSE is used to evaluate the precision of the forecasts, comparing the true x2 and x3 components to the estimated ones, whereas the coverage probability is used to evaluate the reliability of the prediction, evaluating the proportion of true trajectories falling within**

the 50% prediction interval of x2 and x3. Examples of predictions are given in Fig. 4. It shows bad linear predictions of the model with only [x2, x3] (dashed black lines). As the M operator is not time-dependent, the predictions are quite similar, close to the persistence. Then, adding one (green) or two (red) hidden components in the M operators creates some nonlinearities in the forecasts.

In Fig. 5, the predictions are evaluated over the whole test dataset, for different lead times. By introducing hidden components, the RMSE decreases for both x2 and x3 components (top panels). For instance, for a lead time of 0.05, when considering two hidden components, the RMSE is halved when it is compared to the naive linear model without hidden components. The coverage probability metric is also largely improved (bottom panels). Indeed, the results with two hidden components are close to 50%, the optimal value.

To evaluate where the linear model with [x2, x3, z1, z2] performs better than the one with [x2, x3], the Euclidean distances between the forecasts (for a lead time of 0.1) and the truth are computed. Those errors are evaluated at each time step of the test dataset, in the (x2, x3) space. Based on those errors, Fig. 6 shows the relative improvement between the model without and the model with hidden components. When the two models have similar performance, values are close to 0 (white), and when the model including z1 and z2 is better, values are close to 1 (red). Figure 6 clearly shows that error reduction is not homogeneous in the attractor. The improvement is moderate in the outside of the wings of the attractor, but important in the wing-transition. It suggests that the introduction of the hidden components z1 and z2 makes it possible to provide information on the position in the attractor and thus to make better predictions, especially in bifurcation regions.”

Additional questions about the experiments:

1. ‘10 loops of the Lorenz-63 system’ (L 104-105) Do you mean 10 model time units or 10 revolutions on the model attractor? In any case, I would not say that this is a small period of time, compared to the doubling time which is 0.78 MTU.

**Thanks for the suggestion, it now reads l. 133: “10 model time units of the Lorenz-63 system” and we removed “a small period of time”.**

2. From what I understand (L 104-106), you have access to the true x2 and x3 (no observation noise) every dt = 0.001 (which is probably the integration time step for the truth). This seems to be very strong requirements. Can you discuss this?

**Yes, this is a strong requirement, but for the sake of simplicity, we decided to keep dt=0.001, showing that only two latent variables are needed. It is now explained in l. 144: “Note also that the methodology has been tested with larger dt (i.e., 0.01 and 0.1). The conclusion is that by increasing the time delay between observations, it significantly increases the number of latent variables (results not shown).”**

3. What is the choice of the data assimilation window length for the ensemble Kalman smoother? Without further details, I assume that it covers the entire experiment, i.e. $10^4$ observation steps. This is really huge. Can you discuss this?

**L. 146, it is mentioned that: "Finally, the assimilation window length corresponds to $10^4$ time steps. By reducing this length (e.g., to $10^3$, $10^2$, $10^1$), the conclusions remain the same as for dt=0.001."**

Technical comments and suggestions

L 17-18 'using Bayesian framework' → 'using a Bayesian framework' ?

**Done.**

L 21 'All the approaches cited above are assuming that the full state of the system is observed' This is not true: at least Tandeo et al. (2015), Lguensat et al. (2017), Bocquet et al. (2019), Brajard et al. (2020), Fablet et al. (2021) use sparse observation operators in their methods. I would replace 'All the approaches cited above' by 'Many approaches'.

**Done.**

L 23-24 'To deal with those strong constraints' I would replace here 'constraints' by 'assumptions' in order to avoid a potential confusion with strong-constraint methods in variational data assimilation.

**Done.**

L 24-26 'An option is to [...] whereas an other option is to [...]' I would suggest to also mention here the combination of data assimilation and machine learning, because (i) this is what is used in some of the previously cited papers (the ones that can handle sparse and noisy observations), and (ii) this is what is used in the present manuscript!

**Thanks, this sentence now reads, l. 37: "To deal with those strong constraints, i.e., when the model is unknown and when some components of the system are never observed, combination of data assimilation and machine learning shows potential (see e.g., Wikner et al. 2021)."**

L 29 'with a dynamical model (model- or data-driven)' I would replace here 'model-driven' by 'based on physical knowledge' or something like this (to avoid a model-driven model).

**We replaced "model-driven" by "physics-driven", this seems to be the adequate term.**

L 31 'estimation of the parameters' Which parameters?

**It is now clarified, l. 46: "dynamical parameters", i.e. M and Q matrices in our case.**

L 40-41 'from data assimilation, machine learning, and theory of dynamical systems'
→ 'from data assimilation, machine learning, and dynamical systems' ?

**Done.**

L 42 'from partial observations y' In data assimilation, observations are usually noisy in addition to being partial.

**We added, l. 66: "and noisy".**

L 42-46 In this paragraph, why didn't you mention the crucial role of the background error statistics?

**We added, l. 70: "But this estimation requires good estimates of model and observations error statistics (see e.g., Dreano et al., 2017; Pulido et al., 2018)."**

L 48 'to mathematically approximate the dynamic of the system'
→ 'to mathematically approximate the system dynamics'.

**Done.**

L 76 In equation (2), I would suggest to explicit the definition of L, i.e. use something like that:

$$\mathcal{L} \triangleq p(\mathbf{y}_1, \ldots \mathbf{y}_T | \mathbf{x}_1^f, \ldots \mathbf{y}_T^f) \propto \prod_{t=1}^{T} \cdots$$

**Thanks, the mathematical definition of the likelihood has been introduced in Eq. (2), l. 104.**

Furthermore, T is undefined in this equation.

**L. 103, we added: "is computed using T time steps such that".**

L 78-79 'The innovation likelihood given in Eq. (2) is interesting because it corresponds to the squared distance between the observations and the forecast normalized by their uncertainties, represented by the covariance $\Sigma t$.' In data assimilation, this quantity is simply called 'the likelihood'.

**We prefer to keep "innovation likelihood" because different likelihoods appear in DA: the likelihood of the innovation and the total likelihood of the state-space model (see Tandeo et al. 2020, section 4, available here: https://tandeo.files.wordpress.com/2020/11/tandeo_2020_mwr.pdf).**

L 89-90 'This random sampling is used to exploit the correlations between the components of the state vector' I do not understand why this is necessary. Could you elaborate?

**Sorry, it was not clear. It now reads, l. 117: "This random sampling is used to exploit the linear correlations between the components of the state vector, which appear in the non-diagonal terms of P^s."**

L 110 'After 30 iterations of the algorithm presented in section 2, the hidden component z1 is stabilized.' Can you please explain the exact meaning of 'stabilized' in this context?

**In l. 138, we replaced "is stabilized" by "has converged".**

L 114 'this augmented state procedure is repeated' → 'this state augmentation process is repeated'.

**We prefer, l. 141: "this state augmentation procedure is repeated".**

L 117 'z3 is very flat' I would replace 'very' by 'rather' in this statement.

**We prefer, l. 151: "z3 remains close to 0".**

L 125 'Finally, the inclusion of z3 reduces the likelihood (purple lines).' Do you have an explanation for this phenomenon?

**Thanks for this question. After investigation, we discovered that, l. 162: "Finally, due to a significant increase of the forecast covariance $P^f$ in Eq. (2), the inclusion of z3 reduces the likelihood (purple lines). This suggests that a third variable is not needed, and is even detrimental to the skill of the reconstruction."**

L 131 In equation (6), I would explicit the dependence on time, i.e. replace dist (M) by dist(M)(t).

**We hope that the new Eqs. (7) clarify this point.**

L 137-138 'Are they correlated with the unobserved component x1 or with the observed one x2 and x3?' → 'Are they correlated to the unobserved component x1 or to the observed ones x2 and x3?' ?

**Done.**

L 139 'It has been found that...' How did you come up with this? As it is presented, it looks like something pulled out of a hat.

**Thank you very much for this remark. It now reads, l. 169: "Using symbolic regression (i.e., using basic mathematical transformations of x2 and x3 to explain z1 and z2), it has been found that the hidden components z correspond to linear combinations of the derivatives of the observations such that: Eqs. (6)". Sorry for this important omission.**

L 47-48 'This is illustrated in Fig. (3), with 50 independent realizations of the proposed algorithm.' Strictly speaking, this is not the case since a and b are not represented in this figure.

**We decided to add a subfigure in the right panel of Fig. 3.**

[Figure]

The caption of Fig. 3 now reads: "Likelihoods as a function of the iteration of the augmented Kalman procedure (left) and estimation of the a2 and a3 parameters (right). Different dynamical models are considered, from none to three hidden components in z, whereas only x2 and x3 are observed in the Lorenz-63 model. The likelihood of 50 independent realizations of the iterative and augmented Kalman procedure are shown."

L 152-153 'Then, when considering z1 and z2 (red lines), the 50 independent realizations reach the same likelihood after 30 iterations.' What about a and b? Are they similar over the 50 realisations?

**Based on the new Fig. 3 (right panel), it now reads, l. 184: "Interestingly, the scatter plot between a2 and a3 shows a circular relationship. This is also the case for b2 and b3 (results not shown)."**

L 153-154 'it will then focuses' → 'it will then focus'.

**Done.**

L 175-176 'the dynamical evolution of the system is retrieved with our methodology'. This is not clearly shown in the experiments.

**We hope that the new Fig. 4 and Fig. 5 give more information about this. However, the sentence was maybe too strong and we replaced it, l. 236, by: "the dynamical evolution of the system is relatively well captured, for short lead times, with our methodology."**

---

## Author Comment (AC2)

**Response to Reviewer 2**

Data-Driven Reconstruction of Partially Observed Dynamical Systems, by Tandeo et al.

**Note: your comments and questions are reported in this document and we use bold text for our responses.**

This work presents a data-driven method to infer a linear stochastic model from a partially observed system. This work is well-written and contains interesting parts, especially the not-so-common effort to explain the dynamics of the latent (embedding) space. Nevertheless simplifications made in the work do reduce a lot the impact of this paper. Also, there is very little novelty in the approach. The principle of alternating between DA and a data-driven model has been already applied, in more challenging settings (noisy/sparse observations, model with more dimensions). The fact to have a variable that is never-observed has also already been tested. The originality of the approach to have a stochastic model and to explain the latent space is not very developed.

**Thank you for your general comment. Indeed, the combination of data assimilation and machine learning is not new. However, the introduction of latent variables in this context is new (to the best of our knowledge). This is the key methodological contribution of our manuscript.**

**As the Reviewer mentioned, the explanation of the latent space and the stochasticity of the model are the main points of our work. In the new version of the manuscript, these points are discussed in more detail, especially by the addition of 3 new figures: Fig. 3 (right panel), Fig. 4, and Fig. 5.**

Other general comments:
- the justification of the setting and the approach is not convincing to me (see my comments about the abstract and the introduction), and I fail to foresee the real application of the approach. Maybe rephrasing the last part of the conclusion and putting it in the introduction instead could help regarding that matter.

**Thank you very much for your suggestion. We followed the Reviewer's advice, by rephrasing the end of the conclusion and putting it in the introduction, l. 13: "In geophysics, even if one has the perfect knowledge of the studied dynamical system, it remains difficult to predict because of the existence of nonlinear processes (Lorenz, 1963). Beyond this important difficulty, achieving this perfect knowledge of the system is often impossible. Consequently, the governing differential equations are often not known in full because of their complexity, in particular regarding scale-interactions (e.g., turbulent closures are often assumed rather than "known" per se). On top of these two major difficulties, the state of the system is not and cannot be exhaustively observed. Potentially crucial components are and might remain partly or fully out of reach of proper monitoring (e.g., deep ocean or small scale features).**

**Predicting a partially observed and partially known system is therefore a key issue in current geophysics and in particular for ocean, climate and atmospheric sciences."**

- The data-driven model used is linear. It is acknowledged by the authors in the conclusion, but it is one limit of the approach. Maybe the linear approach works because the setting is simple enough (low dimension, weakly non-linear). But also, I wonder if the interpretability of the latent space is precisely related to the choice of the linear model (maybe with a non-linear model, there is no need for a latent space to emulate observed variables...)

**This is an important remark and a discussion is now given in l. 52: "The proposed methodology is based on an important assumption: the surrogate model is linear. Although it can be considered as a disadvantage compared to nonlinear models, this linear assumption also has interesting properties. Indeed, nonlinear model combined with state-augmentation is a very broad family of model and may lead to identifiability issues. Using a linear dynamics already leads to a very flexible family of model since the latent variable may describe nonlinearities and include for example any transformation of the observed or non-observed components of a dynamical model. Furthermore, it allows a rigorous estimation of the parameters using well established statistical algorithms which can be run at a low computational cost."**

**We also added a perspective of work, l. 244, related to this comment: "In future works, we plan to compare the global and local linear approaches (i.e., fix or adaptive linear surrogate model). We also plan to compare them to nonlinear surrogate models, based on neural network architectures with latent information encoded in an augmented space or in hidden layers (e.g., LSTM)."**

- The experiment is done on the Lorenz 63 model, which is very low-dimensional (3) and weakly non-linear. See for example: https://raspstephan.github.io/blog/lorenz-96-is-too-easy/# There are toy models (L96, QG) that could display more interesting behaviors for this methodology.

**Thank you very much for this proposition. We added this clarification in l. 58: "The proposed methodology is evaluated on a low-dimensional and weakly nonlinear chaotic model. As this paper is a proof of concept, a linear surrogate model is certainly well suited for this situation."**

**We added some perspective of work, l. 247: "In this paper, we have demonstrated the feasibility of the method on an idealized and comprehensive problem, using the Lorenz-63 system. In the future, we plan to apply the methodology to more challenging problems, like the Lorenz-96 system or a quasi-geostrophic model. For the application on real data, we plan to use a database of observed climate indices and try to find latent variables that help to make data-driven predictions."**

- The forecast is evaluated only in the next time step, which is again a very easy case. How would behave the forecast over several time steps?

**Thanks for your remark. We have added a new Fig. 4 showing statistical forecasts over several time steps.**

[Figure]

The caption of new Fig. 4 reads: "Example of three statistical forecasts of x2 (left) and x3 (left) with their 50% prediction interval using 3 different linear operators with: no hidden component (dashed black), one hidden component (green), and two hidden components (red). These predictions are obtained using sequential statistical forecasts, as explained in Eqs. (8), on an independent test dataset."

Then, a discussion related to this Fig. 4 has been added, l. 200: "Examples of predictions are given in Fig. 4. It shows bad linear predictions of the model with only [x2, x3] (dashed black lines). As the M operator is not time-dependent, the predictions are quite similar, close to the persistence. Then, adding one (green) or two (red) hidden components in the M operators creates some nonlinearities in the forecasts."

- The method is interestingly stochastic, but no ensemble metrics are used to evaluate the work which would have been interesting.

We have also introduced a new Fig. 5, showing the evolution of the RMSE and the coverage probability, a simple metric to evaluate the estimated prediction intervals.

[Figure]

**The caption of new Fig. 5 reads: "Root Mean Square Error (top) and 50% coverage probability (bottom) as a function of the lead time (x-axis) for the reconstruction of the components x2 (left) and x3 (right). These metrics are evaluated on an independent test dataset."**

**Then, a discussion related to this Fig. 5 has been added, l. 204: "In Fig. 5, the predictions are evaluated over the whole test dataset, for different lead times. By introducing hidden components, the RMSE decreases for both x2 and x3 components (top panels). For instance, for a lead time of 0.05, when considering two hidden components, the RMSE is halved when it is compared to the naive linear model without hidden components. The coverage probability metric is also largely improved (bottom panels). Indeed, the results with two hidden components are close to 50%, the optimal value."**

Specific comments:

Abstract: The 2 first sentences of the abstract is a justification of the approach. Due to the limited size of the abstract, this justification cannot be extended making it too simplistic: 1) It is true that defining a set of equations is difficult, but I would say that a bigger issue is the resolution of the existing set of equations given that coefficients are unknowns and that a discretization is needed for the numerical resolution, which introduces some errors. 2) If we follow the narrative, it is well justified that we should cope with "imperfect equations". But here, the choice is to assume that no equation is known. The fact that those two points are overlooked makes the narrative a bit too simple to be convincing for me. I would suggest starting right away with what you want to achieve in the abstract and having an extended justification in the introduction.

**Thank you for your comment, but we want to keep these first sentences at the beginning of the summary. Although they are overly simplistic, we believe they provide important context for this study. In addition, the summary with these sentences is 198 words, which does not exceed the limit of 200 words. However, we agree with your comments and have taken them into account in the new version of the introduction, between l. 13 and l. 20.**

L13: "governing differential equations are not necessarily known": I would like to see examples of that. I think that, even if some equations are known, a fully data-driven system can be justified, but here, this core question is eluded: What is the range of applications of a purely data-driven model from partial observations?

**To develop this point, we added the following paragraph, l. 21: "A typical example of such a framework is the use of climate indices (e.g., Global Mean Temperature, Niño 3.4 index, North Atlantic Oscillation index) and the study of their links and their dynamics. In this context, the direct relationship between those indices is unknown, even if their more indirect and complex relation exist, through the full knowledge of the climate dynamics. Also, it is highly possible that climate indices are dependent on components of the climate that are not currently considered as key indices, and so are not fully monitored. However, these key indices could be sufficient to describe the most important aspect of climate, leading to accurate and reliable predictions, and enabling cost-effective adaptation and mitigation."**

L21: "All the approaches cited above are assuming that the full state of the system is observed, which is a strong assumption." This is misleading. The papers above (at least Fablet, Bocquet and Brajard) assume that observations are noisy and sparse, but indeed each variable has a non-null probability to be observed. Is it what you mean by "the full state is observed?" There are also many works done in the case a variable is never observed, e.g.: https://arxiv.org/pdf/2102.07819.pdf

**This mistake has been also detected by Reviewer 1. Based on his/her suggestion, we replaced "all" by "many" in l. 35. Moreover, we added a reference to the paper Wikner et al. 2021, l. 37: "To deal with those strong constraints, i.e., when the model is unknown and when some components of the system are never observed, combination of data assimilation and machine learning seems relevant (see e.g., Wikner et al. 2021)." Thanks for the suggestion.**

Figure 1. My understanding is that the paper aims at going one step forward into learning a data-driven model from a realistic setting (by assuming that the state is not fully observed), but it assumes later on that a part of the state is always observed with a very small error. To me, this is a very strong assumption, even stronger than assumptions made by the existing cited papers. So again, I don't see what application is targeted by this work.

**We clarified this point, l. 77: "Nonlinear and adaptive operators as well as noisy observations could be taken into account but, for the sake of simplicity, only the linear and non-noisy case is considered in this paper." The objective of this paper is to retrieve dynamical information that is never observed. But the observed data is assumed to be of good quality (i.e., not noisy).**

L110: the "sequential methodology": Is there a theoretical reason to add sequentially the hidden components or is this mainly practical? How do you see that applied with high-dimensional systems in which, e.g., 10^5 variables are non-observed?

**Indeed it is possible to add all the latent variables at the same time. This is now clarified in l. 142: "Note that several hidden components can be added all at once (results not shown), with similar performance as the sequential procedure described above. In this all at once case, the interpretation of the retrieved components is not as informative, thus we focus on the sequential case." We decided to retain the iterative strategy, especially to introduce and explain clearly Eq. (6a) and (6b).**

**Regarding the application to high-dimensional systems, we refer to the perspective of work, between l. 247 and l. 250.**

L140: This part is, in my opinion, the most interesting part. But I miss some details to fully understand what is done (see below)

**Your next comment is related to this one. See my response below.**

Eq 7: How do you derive those equations? Is it by trials/error or is does it correspond to theoretical reasons?

**Thank you very much for this remark. It now reads, l. 169: "Using symbolic regression (i.e., using basic mathematical transformations of x2 and x3 as regressors to explain z1 and z2), it has been found that the hidden components z correspond to linear combinations of the derivatives of the observations such that: Eqs. (6)". Sorry for this important omission.**

L150 "correspond to $a3 \approx 0$ and to $a2 \approx 0$, respectively": sorry I don't get the "respectively" here, in which case a3 is 0 and in which case a2 is 0?

**The explanation is easier to follow with the new Fig. 3 (right panel). It now reads, l. 181: "As shown in Fig. 3 (right panel), the minimum likelihood corresponds to a3 = 0 and the maximum likelihood corresponds to a2 = 0."**

L150-151: "This suggests that xdot3 is more important than xdot2": Why is that? you still have b2 coefficient associated with xdot2…

**To better understand this sentence, we added l. 182: "Thus, the likelihood when z1 = a3 ẋ3 is higher than when z1 = a2 ẋ2." This is why it is said that ẋ3 is more important than ẋ2 in the reconstruction of the latent variables.**

L153-155: sorry I have read this part several times, and I still don't understand. What does it mean that "the algorithm focuses on the estimation of a_2" I don't see where is the estimation of a_2 in the algorithm and I don't understand what is meant by "focus".

**Sorry for that. We agree that it was not ideally explained. We rephrased it, l. 186: "It means that if a3 = 0 when considering only z1, then b3 ≠ 0 when introducing z2. In terms of forecast performance, this is similar to a2 = 0 and b2 ≠ 0, because the likelihoods converge to the same value (red lines after 30 iterations)."**

L158: The term "model-driven" is misleading. The data-driven model is also a model.

**"Model-driven" has been replaced by "physics-driven" everywhere.**

L175: "the dynamical evolution of the system is retrieved with our methodology. "This is a strong assertion since by construction the evolution of x2 and x3 are observed and you test the forecast skill over only one time-step.

**We hope that the new Fig. 4 and Fig. 5 give more information about this. However, the sentence was maybe too strong and has been replaced, l. 236 by: "the dynamical evolution of the system is relatively well captured, for short lead times, with our methodology."**

End of the introduction: I think it would be nice to have part of these comments in the introduction, justify the approach.

**Indeed, it is better to give the context in the introduction. The beginning of the introduction has been entirely rewritten, between l. 13 and l. 26.**